# Teasing and Internet Harassment among Adolescents: The Mediating Role of Envy and the Moderating Role of the Zhong-Yong Thinking Style

**DOI:** 10.3390/ijerph19095501

**Published:** 2022-05-01

**Authors:** Qiong Wang, Ruilin Tu, Yihe Jiang, Wei Hu, Xiao Luo

**Affiliations:** 1School of Education, Zhengzhou University, Zhengzhou 450001, China; wangq@zzu.edu.cn (Q.W.); 202012122012109@gs.zzu.edu.cn (R.T.); 202012122012108@gs.zzu.edu.cn (X.L.); 2Shanghai Mental Health Center, Shanghai Jiao Tong University School of Medicine, Shanghai 200030, China; jiangyihe@sjtu.edu.cn; 3Information Engineering University, Zhengzhou 450001, China

**Keywords:** weight-related teasing, competency-related teasing, Internet harassment, envy, Zhong-Yong thinking style

## Abstract

Extant research studies have documented that teen victims of traditional bullying are more likely to be perpetrators of Internet harassment. Teasing is a typical form of verbal bullying, but, unfortunately, its link with Internet harassment perpetration has rarely been investigated to date. Therefore, this study constructed two moderated mediation models to explore the relation between weight-related/competency-related teasing (the two main types of teasing) and Internet harassment, and whether these associations are mediated by envy and moderated by the Zhong-Yong thinking style. A total of 1811 Chinese adolescents (*M_age_* = 13.22 years, *SD* = 0.97) were investigated to examine the two models. The results clarified that: (1) weight-related/competency-related teasing was positively predictive of adolescent Internet harassment; (2) envy acted a partially mediating role in the associations between weight-related/competency-related teasing and Internet harassment; (3) the Zhong-Yong thinking style not only moderated the link between weight-related/competency-related teasing and envy, with the effect being more profound for a high-level Zhong-Yong thinking style possessed by adolescents, but also moderated the direct link between competency-related teasing and Internet harassment, with the relation being more potent when the level of Zhong-Yong thinking style was low. Nevertheless, the direct link between weight-related teasing and Internet harassment was not moderated by the Zhong-Yong thinking style. These findings are important to comprehend the psychological mechanisms linking weight-related/competency-related teasing to Internet harassment, and provide some enlightenment for preventing and intervening in adolescent Internet harassment perpetration.

## 1. Introduction

With the widespread use of the Internet, the Internet penetration rate of minors reached 93.1% in 2019 in China (CNNIC, 2020). The use of the Internet can bring many benefits to individuals, but just as any coin has two sides, it can also be a source of potential harm, such as Internet harassment, which is defined as “an overt, intentional act of aggression towards another person online” [1]. In recent years, Internet harassment has become a serious issue and the occurrence rate among adolescents is on the rise [2,3]. Previous studies have indicated that Internet harassment can lead to many negative consequences for both victims and perpetrators, such as anxiety, depression, and suicidal tendencies [4,5,6]. Given the growing rate and negative consequences, it is urgent to explore the influencing factors and mechanisms of Internet harassment.

Extant research studies have documented that the victims of traditional bullying are more likely to become Internet harassment perpetrators [7,8,9]. Traditional bullying can take many forms: physical (e.g., hitting), verbal (e.g., teasing), and social (e.g., social exclusion) [10]. Remarkably, verbal bullying has been shown to have a high prevalence, peaking in secondary school and remaining relatively high in senior high school [11]. Teasing as a typical form of verbal bullying refers to deliberate provocation accompanied by joking off-record remarks that are associated with the target of the tease [12]. Unfortunately, the link between teasing and Internet harassment perpetration has rarely been investigated to date. As a result, this study has the purpose of exploring the influence of teasing on adolescent Internet harassment perpetration and ultimately probes into the potential psychological mechanisms behind this association.

### 1.1. Teasing and Internet Harassment

As a common form of verbal bullying that relies on negative commentary, teasing is becoming an increasingly prevalent problem among Chinese adolescents [13] and has been shown to be associated with adverse health outcomes [14,15]. According to the Frustration–Aggression Theory, individuals who have experienced the negative effects of frustration have strong aggressive inclinations [16]. Suffering from teasing is frustrating for adolescents, which makes them more aggressive. Previous research has also shown that teasing is strongly related to aggressive behavior among adolescents [17]. However, adolescents often disguise their aggressive inclinations in real life. There are two main reasons for it. First, the victims of teasing are more likely to make concessions to maintain interpersonal relationships. Second, the two distinct roles involved in traditional bullying (perpetrator and victim) are fairly stable [18]. The victims of teasing cannot change the situation easily in real life.

Compared with the real world, the virtual world facilitates the victims of teasing to vent their aggressive inclinations. The Internet offers the potential for sending anonymous messages. Adolescents can attack others through the Internet, while hiding their identity, so that the possibility of punishment is reduced [19]. Therefore, it is possible for the vulnerable victims to perpetrate Internet harassment. Combined with the preceding theoretical and empirical grounds, we postulate the first hypothesis:

**Hypothesis** **1** **(H1).***Teasing is positively correlated with Internet harassment*.

### 1.2. The Mediating Role of Envy

To comprehend the “how” mechanism behind the relationship between teasing and Internet harassment, we investigated the mediating effect of envy on this association. Envy generally refers to a bitter emotion characterized by feelings of inferiority, hostility, and resentment, usually caused by a lack of the good qualities, achievements, or possessions of others in social comparisons [20]. The General Strain Theory (GST) can be used as a theoretical guide to understand the mediating role of envy [21]. The GST holds that stress elicits negative affective states, and thus, the possibility of delinquent behaviors to vent the effects of negative emotions increase greatly. Agnew listed three main sources of stress, one of them being noxious situations or events [21], which is similar to the experience of being bullied [22,23]. Empirical research has demonstrated that anger, a kind of negative emotion, plays a mediating role between cyber victimization and cyberbullying [24]. Noteworthily, in addition to anger, individuals who experience bullying victimization also feel other negative emotions, such as envy.

On the one hand, envy may be an outcome influenced by teasing. Teasing focuses on the victim’s poor body shape or competence [25], implying social comparisons with others and highlighting the victim is at a disadvantage. When individuals feel someone else is better than them through a social comparison, they experience envy [26,27]. Therefore, adolescents who have a history of being teased feel envy. Empirical researches are in support of this notion. For example, studies have shown that teasing is linked to a decrease in self-esteem [28], while self-esteem can negatively predict envy [29,30].

On the other hand, envy may predict more Internet harassment perpetration. Envy is a hostile emotion, usually eliciting aggressive behavior [27]. When an individual envies others, they are inclined to act in a malicious way [31]. Results from previous researches have shown that envy can predict the expression of verbal aggressive behaviors [32]. Given the anonymity in cyberspace and convenience of the Internet, the victims of teasing are more prone to engage in Internet harassment perpetration as the externalized response to envy. Empirical studies are in line with this notion. For instance, research has found envy predicts cyberbullying positively among adolescents [33]. Combined with the preceding theoretical and empirical grounds, we propose the second hypothesis:

**Hypothesis** **2** **(H2).***Envy mediates the association between teasing and Internet harassment*.

### 1.3. The Moderating Role of the Zhong-Yong Thinking Style

Although teasing may be significantly associated with envy and Internet harassment, it is unlikely that all teasing victims experience envy and exhibit Internet harassment perpetration in the same way. Thus, exploring the possible moderating variables that perhaps affect the relation between teasing and envy as well as the relation between teasing and Internet harassment is crucial. Culture can influence styles of thought, which in turn affects individuals’ characteristics of emotion as well as behavior. In the context of the Chinese culture, the Zhong-Yong thinking style is a traditional and the most influential style of thought [34]. Thus, we introduced the Zhong-Yong thinking style to be the moderator.

As an inherent way of thinking rooted in the heart of the Chinese people, the Zhong-Yong thinking style refers to considering the same thing from multiple perspectives and choosing to act in an appropriate way that takes into account the self and the overall situation, containing three dimensions: multiple thinking, holism, and harmoniousness [35]. To be specific, multiple thinking assesses the degree to think about issues from different perspectives simultaneously when making decisions. Holism requires individuals to integrate external situational information and internal thought. Harmoniousness measures the tendency to behave in a harmonious way after considering the consequences of one’s behaviors. When individuals are teased by others, those with high levels of Zhong-Yong thinking are more inclined to consider a variety of factors in social comparisons, rather than focusing solely on the past teasing experiences. In addition, Zhong-Yong thinking is an importantly rational cognitive process for emotional regulation [36], which facilitates the reduction in the negative emotional experience of envy. Thus, it is justified to suppose that the Zhong-Yong thinking style is able to mitigate the link between teasing and envy.

Moreover, Zhong-Yong thinking is beneficial to interpersonal relationships and the ultimate goal is to maintain interpersonal harmony [35,37]. Victims of teasing with high levels of Zhong-Yong thinking are more inclined to be aware of the adverse consequences contributed by Internet harassment, emphasize self-restraint, and ultimately choose appropriate behaviors to maintain harmony [38]. A cross-sectional study has revealed that individuals that have high levels of Zhong-Yong thinking lean towards restraining their aggressive impulses and perpetrating less cyberbullying [39]. That is, the higher the Zhong-Yong thinking, the lower the Internet harassment perpetration. Hence, we speculated that the Zhong-Yong thinking style can mitigate the association between teasing and Internet harassment. In conjunction with the preceding theoretical and empirical grounds, we posit the following hypotheses:

**Hypothesis** **3** **(H3).***The association between teasing and envy is**moderated by the Zhong-Yong thinking style, such that the link will be strengthened when Zhong-Yong thinking is low, rather than high*.

**Hypothesis** **4** **(H4).**
*The direct association between teasing and Internet harassment is moderated by the Zhong-Yong thinking style, such that the link will be weakened when the Zhong-Yong thinking style is high, instead of low.*


### 1.4. The Current Study

The purpose of the current research is threefold: firstly, testing whether teasing is predictive of engagement in Internet harassment; secondly, examining the mediating role played by envy in the association between teasing and Internet harassment; and thirdly, investigating the moderating role performed by the Zhong-Yong thinking style in the relation between teasing and envy as well as in the direct relation between teasing and Internet harassment. Previous studies have indicated that teasing consists of two main types. One is weight-related teasing, concentrating on weight and body shape, and the other is competency-related teasing, which concentrates on an individual’s ability [40]. Due to the exploratory nature of this study, two moderated mediation models were constructed, with weight-related teasing and competency-related teasing as the independent variables, envy as the mediating variable, and the Zhong-Yong thinking style as the moderating variable (Figure 1). We expect that differences will exist between the two models.

## 2. Methods

### 2.1. Participants

Our sample was recruited from a junior high school in the Henan province of China using cluster random sampling. A total of 1858 secondary school students participated in our survey. After discarding the invalid questionnaires (i.e., responses with regular answers), 1811 valid questionnaires were obtained, and the valid response rate was 97.47%. In this sample, the age of the adolescents ranged from 11 to 16 years old, with a mean age of 13.22 years (*SD* = 0.97). The participants consisted of 886 females (48.92%) and 925 males (51.08%); 715 (39.48%) were in grade seven, 599 (33.08%) were in grade eight, and 497 (27.44%) were in grade nine.

### 2.2. Measures

#### 2.2.1. Teasing

The Perception of Teasing Scale was used to assess teasing [41]. The scale comprises 11 items and assesses the two factors of weight-related and competency-related teasing. Weight-related teasing includes 6 items, such as “people made fun of you because you were heavy”. Competency-related teasing includes 5 items, such as “people made fun of you because you were afraid to do something”. Participants were asked to rate the frequency of teasing that they encountered on a 5-point scale ranging from 1 = “Never” to 5 = “Very often”. The final score was the sum of the item scores, with higher scores representing higher perceptions of teasing. In the present study, the total Cronbach’s α was 0.91, and Cronbach’s α for weight-related and competency-related teasing were 0.91 and 0.87, respectively.

#### 2.2.2. Envy

The current study used the Facebook Envy Scale [42] to evaluate the levels of envy experienced by the participants. The questionnaire contained eight items (e.g., “many of my friends have a better life than me”), all of which were graded on a 5-point scale ranging from 1 (Strongly disagree) to 5 (Strongly agree). The item scores were added to obtain the final scores, and the higher the scores, the stronger the envy. Considering Chinese netizens cannot use Facebook, we changed the Facebook context to the context of mainstream social media sites in China (e.g., WeChat, QQ, and Weibo). Empirical research in China demonstrated that this measure has good reliability [43,44]. The Cronbach’s α for the present sample was 0.82.

#### 2.2.3. Internet Harassment

Internet harassment was measured using the adaption of the Internet Harassment Scale [45]. Based on the purpose of the current study, the “received” in the original scale was replaced with “made” to assess the students’ Internet harassment perpetration. This scale comprised 3 items (e.g., “in the past year, I have made offensive or threatening comments to others online”), all of which were rated regarding the frequency on a 5-point scale (1 = “Never”, 5 = “Very often”). Responses across these three items were aggregated, with higher scores reflecting higher levels of Internet harassment. In this study, the Cronbach’s α was 0.85.

#### 2.2.4. Zhong-Yong Thinking Style

The Zhong-Yong Thinking Style Scale, explained by Wu and Lin [35], was used to measure the adolescents’ level of Zhong-Yong thinking. This scale consists of three dimensions, “multiple thinking”, “holism”, and “harmoniousness”, with 13 items (e.g., “in arguments, I take into account conflicting opinions”). Each item was graded on a 5-point scale (1 = “Strongly disagree”, 5 = “Strongly agree”). A higher total score manifested a higher degree of Zhong-Yong thinking. In the current study, Cronbach’s α was 0.92.

### 2.3. Procedure

This survey was permitted by the Ethics Committee of the first author’s institution. Informed consent was obtained from the legal guardians before data collection. Professionally trained psychology graduate students used a unified questionnaire and administered the investigation in class. The instructions were consistent and underlined the principles of confidentiality and anonymity to lower students’ worries. Participants filled in questionnaires regarding demographics, teasing, envy, Internet harassment, and the Zhong-Yong thinking style. After completing the survey, the questionnaires were collected on the spot.

### 2.4. Data Analysis

In this study, we applied SPSS 25.0 (IBM, Armonk, NY, USA) and the Hayes SPSS macro program PROCESS 3.3 (http://www.afhayes.com, accessed on 24 October 2021) to analyze the data. This PROCESS macro for SPSS is specifically designed to examine sophisticated models involving both mediating and moderating variables [46]. The analyses were conducted in three steps. Firstly, we calculated descriptive statistics and correlation analysis for each variable. Secondly, we used the PROCESS macro (Model 4) to test the mediating role played by envy in the relationship between weight-related/competency-related teasing and internet harassment. Finally, we used Model 8 to test the moderated mediation models, investigating the moderating role acted by the Zhong-Yong thinking style. The mediation and moderating effects were examined by using the bias-corrected percentile bootstrap method. A bootstrap sample of 5000 was drawn, obtaining 95% confidence intervals (CI). If the 95% confidence interval excludes zero, it means that the effects are statistically significant. All variables were standardized prior to the formal data processing.

## 3. Results

### 3.1. Preliminary Analysis

Means, standard deviations, and zero correlations for the study variables are shown in Table 1. As expected, both weight-related and competency-related teasing were positively correlated with envy (*r* = 0.26, *p* < 0.001; *r* = 0.42, *p* < 0.001) and Internet harassment (*r* = 0.22, *p* < 0.001; *r* = 0.32, *p* < 0.001). In addition, envy was positively related to Internet harassment (*r* = 0.23, *p* < 0.001). The results of the correlational analysis initially supported the hypotheses.

Notably, we divided the participants into early adolescence (11–13 years old) and middle adolescence (14–16 years old) based on their development trajectory and age range. Then, we conducted an independent-sample t-test for age as well as sex. The results revealed that there were no significant age and gender differences in weight-related teasing, competency-related teasing, and the Zhong-Yong thinking style, whereas both age and sex differences existed in envy and Internet harassment. Specifically, the levels of envy (*t* = 2.25, *p* < 0.05) and Internet harassment (*t* = 2.55, *p* < 0.05) were significantly higher in middle adolescence than in early adolescence. In addition, females reported significantly higher envy scores (*t* = 2.34, *p* < 0.05) and lower Internet harassment scores (*t* = −3.93, *p* < 0.001) than males. Thus, we included sex and age as covariates in subsequent analyses to obtain pure effects.

### 3.2. Testing for Mediation Effect

We utilized Model 4 of the PROCESS macro to examine the mediating effect of envy on the relationship between weight-related/competency-related teasing and Internet harassment. Table 2 presents the results of mediation testing. Standardized regression coefficients manifested that the positively predictive effect of weight-related or competency-related teasing on Internet harassment was significant (Model 1, *β* = 0.22, *p* < 0.001; Model 2, *β* = 0.31, *p* < 0.001). Moreover, both weight-related and competency-related teasing positively predicted envy (Model 1, *β* = 0.26, *p* < 0.001; Model 2, *β* = 0.43, *p* < 0.001), which, in turn, had a significantly and positively predictive effect on Internet harassment (Model 1, *β* = 0.19, *p* < 0.001; Model 2, *β* = 0.12, *p* < 0.001). Furthermore, when the mediating variable was added, the direct effect of weight-related or competency-related teasing on Internet harassment was still significant (Model 1, *β* = 0.17, *p* < 0.001; Model 2, *β* = 0.26, *p* < 0.001). Finally, the bias-corrected percentile bootstrap analyses showed that the direct effect of weight-related or competency-related teasing on Internet harassment and the mediating effect of envy were all significant. To be specific, envy partially mediated the relation between weight-related teasing and Internet harassment in Model 1 (standardized indirect effect = 0.05, SE = 0.01, 95% CI = [0.03, 0.07]), and the mediation effect accounted for 21.73% of the total effect. In addition, in Model 2, envy played a partial mediating role in the link between competency-related teasing and Internet harassment (standardized indirect effect = 0.05, SE = 0.01, 95% CI = [0.02, 0.08]), and the mediation effect accounted for 16.02% of the total effect.

### 3.3. Testing for the Moderated Mediation

To investigate the moderating role of the Zhong-Yong thinking style in the moderated mediation models, we employed Model 8 of the PROCESS macro. The results are displayed in Table 3. In Model 1, the interaction (product term) between weight-related teasing and the Zhong-Yong thinking style significantly and positively affected envy (*β* = 0.05, *p* < 0.01), but did not affect Internet harassment (*β* = 0.003, *p* = 0. 87). In addition, the Model 2 of Table 3 demonstrated that the interaction (product term) between competency-related teasing and the Zhong-Yong thinking style positively predicted envy (*β* = 0.05, *p* < 0.01), while negatively predicted Internet harassment (*β* = −0.12, *p* < 0.001). Thus, the Zhong-Yong thinking style moderated the association between weight-related/competency-related teasing and envy as well as the direct association between competency-related teasing and Internet harassment.

To elaborate the essence of the moderating effect, we conducted simple slope test. As shown in Figure 2 and Figure 3, weight-related or competency-related teasing had a significantly predictive function on envy, regardless of the low levels or high levels of the Zhong-Yong thinking style that are possessed by individuals (M − 1 SD and M + 1 SD, respectively). However, weight-related teasing had a greater impact on envy for participants who had high levels of Zhong-Yong thinking (*β*_simple_ = 0.28, SE = 0.03, 95% CI = [0.22, 0.35]) than for participants with low levels of Zhong-Yong thinking (*β*_simple_ = 0.19, SE = 0.02, 95% CI = [0.14, 0.23]). Similarly, compared to individuals possessing low-level Zhong-Yong thinking (*β*_simple_ = 0.35, SE = 0.02, 95% CI = [0.30, 0.39]), the positive association between competency-related teasing and envy was more profound in individuals who possessed high-level Zhong-Yong thinking (*β*_simple_ = 0.45, SE = 0.03, 95% CI = [0.39, 0.51]).

The simple slope plot of Figure 4 presented that the Zhong-Yong thinking style buffered the link between competency-related teasing and Internet harassment. To be specific, competency-related teasing significantly predicted engagement in Internet harassment when the level of Zhong-Yong thinking was low (*β*_simple_ = 0.32, SE = 0.03, 95% CI = [0.27, 0.37]), whereas this predictive effect was diminished when the level of Zhong-Yong thinking was high (*β*_simple_ = 0.07, SE = 0.04, 95% CI = [0.01, 0.14]).

## 4. Discussion

The bullying experience is common among adolescents, especially weight-related and competency-related teasing, which are two types of teasing [40]. Previous researches have documented that being teased about one’s weight or ability is related to a host of adverse psychological and physical outcomes, such as eating disordered behaviors, low self-esteem, and depression [14,47,48]. However, there is a clear paucity of research on whether teasing about weight or competency is correlated with Internet harassment. Thus, two moderated mediation models were proposed to test whether and how weight-related or competency-related teasing works on Internet harassment, and whether all people are equally influenced by it. The findings of our research indicated that weight-related/competency-related teasing was significantly and positively associated with Internet harassment among Chinese adolescents, while envy partially mediated this relationship. Furthermore, the Zhong-Yong thinking style moderated the relationship between weight-related/competency-related teasing and envy as well as the direct relationship between competency-related teasing and Internet harassment. These results have certain theoretical and practical implications for deepening our understanding of the relationship between weight-related/competency-related teasing and Internet harassment.

It is worth emphasizing that the results of the age difference test revealed that, from early to middle adolescence, individuals had significantly higher levels of envy and committed more Internet harassment. This may be because, as teenagers become older, they are more likely to have access to the Internet and spend more time on social media sites. When youths notice some information in terms of other users’ happy smiles, wonderful life, and excellent achievements on social network sites, they will become envious [42]. Additionally, as more and more people engage in cyber aggression, adolescents using social network sites will inevitably notice and then imitate those deviant behaviors, resulting in Internet harassment. The sex difference test also showed that females suffered more feelings of envy than males, and perpetrate less Internet harassment. These findings are consistent with the notion that significant variability in emotion regulation abilities as well as the severities of Internet harassment perpetration exist among adolescent females versus males [49,50]. On the one hand, compared with males, females are more sensitive to comparisons with others, and more frequently ruminate on their inferiority, then generating more feelings of envy. On the other hand, males have lower empathy than girls, while people with low empathy are more inclined to engage in Internet harassment [51,52].

### 4.1. Teasing and Internet Harassment

Consistent with Hypothesis 1, the present study found that teasing, either weight-related or competency-related teasing, was positively related to Internet harassment. That is to say, adolescents who have experienced teasing are more likely to attack others through the Internet, which is in line with previous studies [8,53]. In addition, the result supported the Frustration–Aggression Theory, which contends that frustrations can create aggressive inclinations, and aggressive behavior may be a psychological defense caused by external pressure situations to alleviate the inner pressure [16]. Since extensive negative verbal commentaries regarding appearance or competency put down them, the experience of being teased is a frustration for adolescents whose concept of self is rapidly developing [13,14]. Due to the development of electronic information and communication technology, allowing for assaulters to remain anonymous and unaccountable for bullying actions [54], teasing victims will become perpetrators of bullying in the cyberspace as a way of obtaining revenge or expressing stress with a low possibility of detection.

### 4.2. The Mediating Role of Envy

As revealed by the results of the current study, envy mediated the relationship between weight-related/competency-related teasing and Internet harassment. Therefore, hypothesis 2 was validated. This mediating effect of envy supported the basic viewpoint of the General Strain Theory (GST), which stresses that negative experiences might create a negative emotional strain, and then turn into bullying behaviors [21]. Namely, negative affective strain caused by teasing victimization, in combination with the anonymity offered by cyberspace, will lead individuals to engage in Internet harassment as an externalization of strain. This finding is consistent with previous research studies that state that negative emotions mediated the association between stressful situations and cyber aggression [24,55].

For the first link between weight-related/competency-related teasing and envy in the mediating model, our study revealed that the relationship is significant and positive, which is in line with previous research suggesting that being teased during childhood or adolescence is associated with a range of emotional problems [56,57]. In adolescence, social feedback from others, such as parents and peers, has long been considered vital to developing and maintaining self-worth [58]. However, adolescents who are teased in terms of weight and ability are apt to have low self-worth as they incorporate negative and even discriminatory evaluations into their concept of self. When making social comparisons, they generally perceive themselves as inferior to others and arise feelings such as “how excellent they are” and “how successful he is”, eventually increasing the risk of envy [26,59].

For the second part of the mediation stage, our study provided evidence for the notion that envy has a significantly predictive effect on Internet harassment perpetration. According to the model of hot/cool system (HCS), which states that people have two behavioral reactive systems: emotional impulse, the “hot system”, and rational cognition, the “cool system” [60]. Envy activates the “hot system”, making individuals more impulsive and not being able suppress aggressive behaviors. In addition, empirical evidence also suggested that envy can result in a greater motivation to harm, as well as a greater likelihood of engaging in harmful behaviors [35,61,62], as the enviers attempt to level the envied person down [63]. Given that the cyberspace provides a comfortable place for the enviers to display their displeasure toward others’ superiority, it is reasonable to find that individuals who envy others are more prone to participate in Internet harassment.

### 4.3. The Moderating Role of the Zhong-Yong Thinking Style

Our results showed that the Zhong-Yong thinking style moderated the relation between weight-related/competency-related teasing and envy as well as the direct relation between competency-related teasing and Internet harassment. However, contrary to our hypothesis 3, the Zhong-Yong thinking style exacerbated the link between weight-related/competency-related teasing and envy, that is, this association was stronger for adolescents with high levels of Zhong-Yong thinking. The result is intriguing and the reasons may be as follows. On the one hand, multiple thinking helps individuals to notice and adopt the opinions of teasers, so they are more likely to form a negative self-image. Moreover, status hierarchy forms during adolescence [64]. Hence, envy may be reinforced among teasing victims with high levels of Zhong-Yong thinking, because it is easier for them to find that they are at a disadvantage in social comparisons. On the other hand, the Zhong-Yong thinking style helps individuals to recognize their inner self-state and external environmental requirements in a holistic manner, and to think and judge in the unity of the opposition between the self and the environment [65]. Victims of teasing with a high level of Zhong-Yong thinking focus on interpersonal harmony and, therefore, choose to restrain their aggressive impulses [39]; envy is not disruptive to their interpersonal harmony, and therefore they tend to cater to their own emotional states when weighing the self and the environment [36]. Previous studies have also found that the higher the levels of Zhong-Yong thinking, the higher the tendency of forgiveness, which is conducive to interpersonal harmony. On the contrary, the higher the levels of Zhong-Yong thinking, the lower the tendency to have self-forgiveness, which does not affect interpersonal harmony [66]. Therefore, it is reasonable to find that the Zhong-Yong thinking style exacerbates the association between weight-related/competency-related teasing and envy.

Regarding our hypothesis 4, the findings indicated the Zhong-Yong thinking style buffered the relationship between competency-related teasing and Internet harassment, while it did not mitigate the relationship between weight-related teasing and Internet harassment. This means that the association between competency-related teasing and Internet harassment was stronger for adolescents with low levels of Zhong-Yong thinking, while adolescents who have been teased about their weight or body shape will perpetrate Internet harassment, regardless of whether they have a high or low level of Zhong-Yong thinking.

On the one hand, competency is multidimensional, and multiple thinking helps individuals to assess their capabilities comprehensively from multiple perspectives. When adolescents are teased about their abilities, such as academic performance or social skills, those holding high levels of Zhong-Yong thinking may not consider other’s negative opinion as a primary view of their competency. In addition, the model of “hot/cool system” (HCS) asserts that individuals will demonstrate more rational and self-control behavior in a “cool system” that is cognition-oriented [56]. Zhong-Yong thinking emphasizing rational cognition will active a “cold system” and allow the individual to respond appropriately, thereby inhibiting aggressive behavior [39]. Hence, competency-related teasing victims with high levels of Zhong-Yong thinking are less harmed by bad comments and choose more appropriate behaviors to respond to teasing, rather than resorting to Internet harassment.

On the other hand, in today’s culture of “thinness is beautiful”, weight has become a sensitive topic and individuals have internalized stereotyped ideal body shapes [13]. Adolescents face an enormously dynamic period in the development of the concept of self, and juvenile victims of weight-related teasing tend to believe they are unattractive for not having an ideal body, forming a negative self-image and a low self-esteem [15,67]. Moreover, empirical research has revealed that weight-related teasing is more frequently experienced by overweight adolescents than youths with a healthy weight [68], and weight is difficult to change in a short period. Thus, the experience of being teased about weight can be a source of social pressure and much concern for youths either possessing high-level or low-level Zhong-Yong thinking. As a result, those victims find it hard to behave rationally and are likely to vent their stress by perpetrating Internet harassment.

### 4.4. Limitations and Implications

Some limitations should be acknowledged in the current study. First, we only used self-reported measures to collect data; thus, social desirability effects or response bias may exist in this work. For example, adolescents are likely to underestimate their Internet harassment perpetration. Future research should improve the objectivity of the data by collecting them from a variety of sources, such as parents, peers, or school administrators. Second, the present study has a cross-sectional design, which means that we cannot infer casual relationships. An experimental or longitudinal design could be adopted for future research to explore the causal relationship among the variables. Third, the representativeness of the sample in this study may be limited, as the participants are all Chinese adolescents. Future studies could be conducted in various cultural contexts as well as comparison studies could be carried out from a cross-cultural perspective. Furthermore, since the focus of the study was the Internet harassment of victims of teasing, future research should consider different groups of victims of bullying, and take into account more complex combinations of online and offline bullying among adolescents. Finally, the results of the independent-sample t-test show that females were more likely to feel envy and perpetrated less Internet harassment compared to males. In addition, teenagers are more inclined to experience envy and commit more Internet harassment with the increase in age. However, we included sex and age as covariates in the analyses to obtain pure mediating and moderating effects. Future studies should explore further the vital role of sex and age in explaining variance in both envy and Internet harassment, or investigate whether sex and age moderate the mediating effect of envy.

Despite these limitations, the current study has some vital theoretical and practical implications. Theoretically, this study, using an integrated model, helps us to understand the unique and interactive effects of weight-related/competency-related teasing, envy, and the Zhong-Yong thinking style on Internet harassment, and supports the Frustration–Aggression Theory, General Strain Theory, and model of “hot/cool system”. It demonstrated that both weight-related and competency-related teasing victimization serve as important influencing factors for an individual’s negative emotions, and subsequently increase their Internet harassment perpetration. Additionally, the current study provides a new perspective for a better understanding of the Zhong-Yong thinking style and the difference between these two types of teasing. Practically, these results contribute to prevent and intervene in adolescent Internet harassment perpetration. First, because those who have a history of being teased about weight and competency are more likely to conduct Internet harassment, school teachers and leaders should pay attention to the mental health of these victims and make early interventions to decrease the likelihood of becoming a perpetrator in the cyberspace. Second, our study clarified the mediation mechanism of envy in the relationship between weight-related/competency-related teasing and Internet harassment; school counselors in the development of prevention and intervention programs can focus on emotion regulation and envy control, not only in the context of real life, but also in the cyberspace. Finally, our study showed that the Zhong-Yong thinking style buffered the relation between competency-related teasing and Internet harassment, but exacerbated the relation between weight-related/competency-related teasing and envy. Hence, for both weight-related and competency-related teasing victims with a high level of Zhong-Yong thinking, school counselors, teachers, and parents should be aware that those adolescents have a stronger negative emotional experience of envy, and should, then, teach them effective emotional management methods to properly control their emotions; for competency-related teasing victims with a low level of Zhong-Yong thinking, corresponding intervention measures to cultivate the Zhong-Yong thinking style can be conducted to control their aggressive behavior in the cyberspace.

## 5. Conclusions

In sum, the current study revealed that weight-related/competency-related teasing victimization is a risk factor for adolescent Internet harassment. In addition, the effect is partially mediated by envy. Moreover, the effect of weight-related/competency-related teasing on envy is stronger for adolescents with higher levels of Zhong-Yong thinking than for lower, while the direct effect of competency-related teasing on Internet harassment is stronger for adolescents with lower levels of Zhong-Yong thinking than for those with higher levels. Nevertheless, Zhong-Yong thinking does not moderate the direct link between weight-related teasing and Internet harassment. These findings provide a deeper understanding of the relationship between weight-related/competency-related teasing and Internet harassment, which contributes to a more detailed prevention and intervention.

## Figures and Tables

**Figure 1 ijerph-19-05501-f001:**
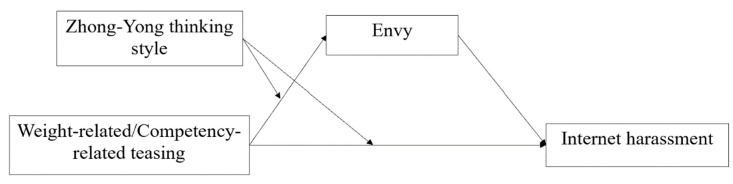
The proposed moderated mediation model.

**Figure 2 ijerph-19-05501-f002:**
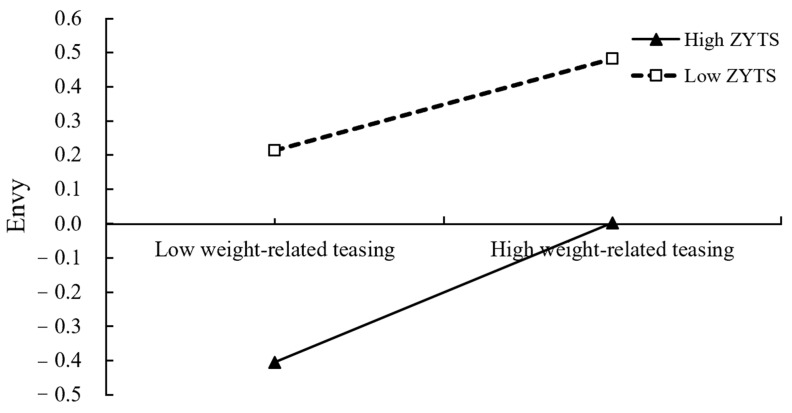
ZYTS as a moderator in the relationship between weight-related teasing and envy (ZYTS = the Zhong-Yong thinking style).

**Figure 3 ijerph-19-05501-f003:**
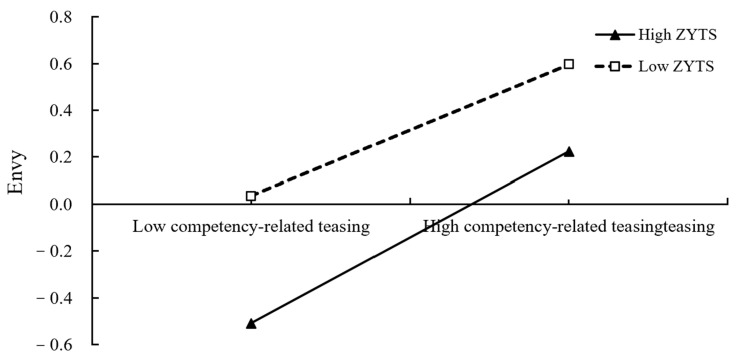
ZYTS as a moderator in the relationship between competency-related teasing and envy (ZYTS = the Zhong-Yong thinking style).

**Figure 4 ijerph-19-05501-f004:**
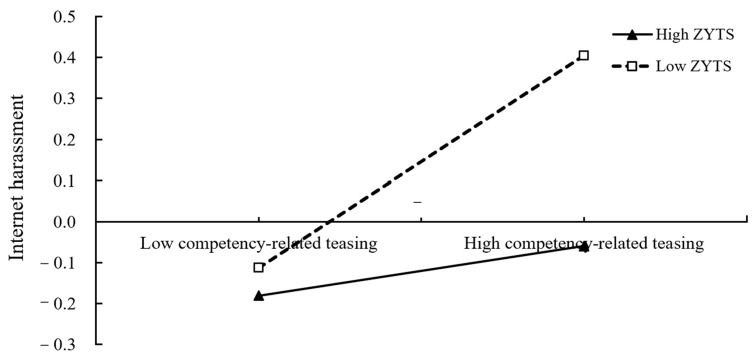
ZYTS as a moderator in the relationship between competency-related teasing and Internet harassment (ZYTS = the Zhong-Yong thinking style).

**Table 1 ijerph-19-05501-t001:** Descriptive statistics and correlations among variables.

	*M*	*SD*	1	2	3	4	5	6	7
1. Sex	0.51	0.50	1						
2. Age	13.22	0.97	0.03	1					
3. Weight-related teasing	7.51	3.44	−0.01	−0.01	1				
4. Competency-related teasing	6.72	2.78	0.04	0.02	0.46 ***	1			
5. Envy	19.20	5.86	−0.06 *	0.09 ***	0.26 ***	0.42 ***	1		
6. Internet harassment	3.33	1.09	0.09 ***	0.07 **	0.22 ***	0.32 ***	0.23 ***	1	
7. Zhong-Yong thinking style	51.65	8.34	−0.02	−0.05	−0.15 ***	−0.20 ***	−0.32 ***	−0.20 ***	1

Note. *n* = 1811. Sex is a dummy variable (female = 0 and male = 1); * *p* < 0.05, ** *p* < 0.01, *** *p* < 0.001.

**Table 2 ijerph-19-05501-t002:** Testing the mediation effect of weight-related/competency-related teasing on Internet harassment.

	(IV)	(DV: Internet Harassment)		(DV: Envy)		(DV: Internet Harassment)	
		*β*	*t*	*β*	*t*	*β*	*t*
Model 1	Sex	0.19	4.06 ***	−0.11	−2.42 *	0.21	4.58 ***
	Age	0.07	2.77 **	0.10	4.31 ***	0.05	2.00 *
	Weight-related teasing	0.22	9.65 ***	0.26	11.43 ***	0.17	7.42 ***
	Envy					0.19	7.93 ***
	*R* ^2^	0.06 ***		0.08 ***		0.09 ***	
	*F*	38.92		51.56		45.93	
Model 2	Sex	0.15	3.45 ***	−0.15	−3.60 ***	0.17	3.86 ***
	Age	0.06	2.58 **	0.09	4.24 ***	0.05	2.11 *
	Competency-related teasing	0.31	14.05 ***	0.43	20.12 ***	0.26	10.73 ***
	Envy					0.12	4.79 ***
	*R* ^2^	0.11		0.19		0.12	
	*F*	74.12 ***		144.10 ***		61.99 ***	

Note. *n* = 1811. IV, independent variable; DV, dependent variable. Sex is a dummy variable (female = 0 and male = 1); * *p* < 0.05, ** *p* < 0.01, *** *p* < 0.001.

**Table 3 ijerph-19-05501-t003:** Testing the moderated mediation effect of weight-related/competency-related teasing on Internet harassment.

	(IV)	(DV: Envy)		(DV: Internet Harassment)	
		*β*	*t*	*β*	*t*
Model 1	Sex	−0.13	−2.90 **	0.20	4.40 ***
	Age	0.09	3.89 ***	0.04	1.92
	Weight-related teasing	0.24	10.30 ***	0.16	6.78 ***
	ZYTS	−0.29	−13.12 ***	−0.12	−5.16 ***
	Weight-related teasing × ZYTS	0.05	2.80 **	0.003	0.16
	Envy			0.15	6.12 ***
	*R* ^2^	0.16 ***		0.11 ***	
	*F*	68.78		35.50	
Model 2	Sex	−0.16	−3.93 ***	0.18	4.04 ***
	Age	0.08	3.85 ***	0.05	2.13 *
	Competency-related teasing	0.40	18.24 ***	0.20	7.79 ***
	ZYTS	−0.24	−11.49 ***	−0.11	−4.79 ***
	Competency-related teasing × ZYTS	0.05	3.18 **	−0.12	−6.93 ***
	Envy			0.10	3.91 ***
	*R* ^2^	0.25 ***		0.16 ***	
	*F*	121.28		55.19	

Note. *n* = 1811. IV, independent variable; DV, dependent variable. ZYTS = Zhong-Yong thinking style. Sex is a dummy variable (female = 0 and male = 1); * *p* < 0.05, ** *p* < 0.01, *** *p* < 0.001.

## Data Availability

The data presented in this study are available upon request from the corresponding author.

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
