# Peer review of "Teasing and Internet Harassment among Adolescents: The Mediating Role of Envy and the Moderating Role of the Zhong-Yong Thinking Style"

_ijerph, 2022, doi:10.3390/ijerph19095501_

Round 1

Reviewer 1 Report

The study showed a very interesting model to investigate the relationship between teasing and Internet harassment. In particular, it explores the role played by envy and the moderating role of Zhong-Yong thinking style. The results highlighted how envy partially mediated this relationship. From a methodological point of view and presentation of the results, the paper is rigorous and in applicative terms, it helps to understand how even cultural belonging can influence styles of thought and in turn affect deviant behaviors. I would only invite the authors to reflect on how these behaviors are also learned in context, for example in technologically mediated one, both in the negative (Paciello, et al. (2021). Online sexist meme and its effects on moral and emotional processes in social media . Computers in human behavior, 116, 106655.) and in a positive sense (D'Errico, F., Leone, G., Schmid, M., & D'Anna, C. (2020). Prosocial virtual reality, empathy, and EEG measures: a pilot study aimed at monitoring emotional processes in intergroup helping behaviors. Applied Sciences, 10 (4), 1196.) This means that in future studies it will be necessary to verify the adolescents' use of social media and new media at least as a control variable, for this study it will be enough to underline it within the limits.

Author Response

Thank you very much for your kind comments on our manuscript (Manuscript reference No. ijerph-1646433). We have revised the manuscript according to the comments and below we address each of the reviewer’s comments with a point-by-point discussion describing the changes we have incorporated.

The following is our response to reviewers’ comments.

The study showed a very interesting model to investigate the relationship between teasing and Internet harassment. In particular, it explores the role played by envy and the moderating role of Zhong-Yong thinking style. The results highlighted how envy partially mediated this relationship. From a methodological point of view and presentation of the results, the paper is rigorous and in applicative terms, it helps to understand how even cultural belonging can influence styles of thought and in turn affect deviant behaviors. I would only invite the authors to reflect on how these behaviors are also learned in context, for example in technologically mediated one, both in the negative (Paciello, et al. (2021). Online sexist meme and its effects on moral and emotional processes in social media. Computers in human behavior, 116, 106655.) and in a positive sense (D'Errico, F., Leone, G., Schmid, M., & D'Anna, C. (2020). Prosocial virtual reality, empathy, and EEG measures: a pilot study aimed at monitoring emotional processes in intergroup helping behaviors. Applied Sciences, 10 (4), 1196.) This means that in future studies it will be necessary to verify the adolescents' use of social media and new media at least as a control variable, for this study it will be enough to underline it within the limits.

Response: Many thanks for your positive and encouraging comments. We reflected on how behaviors or emotions are learned in cyberspace, and our explanations are as follows:

With the widespread use of the internet, the internet penetration rate of minors reached 93.1% by 2019 in China. The use of the internet may bring potential harm to youths, such as envy and internet harassment.

When adolescents notice some information about other users’ happy smiles, enjoyable trips, and personal achievements in social network sites, they will experience envy [1]. In addition, previous indicated Heavy Facebook users tend to feel higher levels of Facebook envy than light Facebook users [2]. That is, the use of social network sites may contribute to envy.

The internet offers the potential for sending anonymous messages. Individuals can attack others to vent their emotions through the internet while hiding their identity, so that the possibility of punishment is reduced [3]. As more and more people engage in cyber aggression, adolescents using social network sites inevitably notice those negative behaviors, and are more likely to imitate, perpetrating internet harassment. Thus, the use of social network sites may increase adolescents’ deviant behaviors.

References:

  1. Lin, R.; Utz, S. The emotional responses of browsing Facebook: Happiness, envy, and the role of tie strength. Computers in Human Behavior 2015, 52, 29-38.
  2. Mehari, K.R.; Farrell, A.D.; Le, A.-T.H. Cyberbullying among adolescents: Measures in search of a construct. Psychology of Violence 2014, 4, 399.
  3. Tandoc Jr, E.C.; Ferrucci, P.; Duffy, M. Facebook use, envy, and depression among college students: Is facebooking depressing? Computers in Human Behavior 2015, 43, 139-146.

We thank you and the reviewers for the comments which make us think and learn a lot, and believe that the incorporation of the amendments makes this manuscript more valuable. We hope the manuscript is now suitable in its present form for publication in International Journal of Environmental Research and Public Health.

Wish you all the best!

Yours sincerely,

Authors

Reviewer 2 Report

Manuscript ID: IJERPH-1646433

Full Title: Teasing and internet harassment among adolescents: The mediating role of envy and the moderating role of Zhong-Yong thinking style

General Comments:

The present study explored the degree to which being teased was predictive of engagement in online harassment, and whether these associations were mediated by envy, and moderated by Zhong-Yong thinking style. The study uses questionnaire data from a large sample of youths, a major strength of the study. However, this is unfortunately where my enthusiasm for the paper dampens. Overall, the introduction and discussion sections offer some good insights but are rather choppy, leaving me wishing there was more fulsome, integrative discussion of the issues at large.  Further, I am considerably worried about the primary outcome measure, harassment. The authors modified an existing questionnaire, which was designed to ask about the degree to which one has been harassed, and asked kids to instead indicate whether they have harassed others. This experience is extraordinarily subjective, and I’m concerned about the validity of such a measure. The tested models are overly simplistic and rely on simple overall mean scores obtained from questionnaires with no dissection of types/severity of teasing, envy, or harassment. Further, the analyses woefully overlook the importance of sex and age in an enormously dynamic period of development spanning the onset of puberty, a time in which these social behaviors are likely most seminal in a person’s life. Finally, the authors interpret extremely small effects as being quite meaningful (e.g., regression coefficients > .10). I urge the authors to consider both statistical and practical significance when interpreting their results. Even the most miniscule effects have a shot at being statistically significant when tested in a sample of 1800+ people, but that doesn’t make them useful or meaningful in the grander scope of understanding human behavior and mental wellness.

Specific Comments:

Introduction:

  1. I think that the intro could use some restructuring for the flow of information. The authors present the goals of the study at the end of the first section without defining several of the key components, e.g., envy and Zhong-Yong thinking style. The next section starts out with a clear definition of another component, teasing, which was only briefly described prior to introducing the study goals. Some basic reorganization of information would make the introduction much clearer and help the justification of the study.

  1. Please Figure 1 – the lines and text look distorted, and there are what appear to be paragraph marks next to the text in each of the boxes.

Methods:

  1. Was there any rationale for the number of participants recruited? E.g., was this part of a larger study, was there any prior power analysis, etc.?

  1. With so many participants, the authors have loads of statistical power to dissect these issues further and explore more nuanced indices of teasing and harassment (e.g., competency versus weight-related teasing). I would like to see more in-depth analyses pulling these issues apart and describing specifics. Lumping together all types of “teasing” and “harassment” is overly ambiguous, and there is no need to do so with such a large N

  1. Could the authors more clearly identify how the Facebook Envy Scale was adapted to reflect Chinese social media sites? Also, how does social media usage/participation/consumption in China compare to that of other nations where social media sites are not state-controlled/limited?

  1. Why did the authors choose to use averages of item scores for each scale as their primary measures, as opposed to other metrics that better capture the variability in scales (e.g., sum scores, factor scores, etc.)?

  1. I’m concerned about the validity of the internet harassment measure in its modified form for the current study. Asking youths if someone has made offensive comments toward them is likely to receive more honesty/accuracy than asking youths if they have said something offensive to another person. Further, perceptions of what is “offensive” versus “a joke/funny” are likely to vary significantly from person to person. Thus, youths engaging in harassment might not answer these questions accurately with respect to the recipient’s perceptions of the situation (e.g., the perpetrator thought it was just a funny joke, whereas the recipient saw it as offensive/hurtful). The authors need to strongly justify and defend the validity of this scale.

  1. Please specify the version of PROCESS used for analyses.

  1. The authors state that they controlled for age and sex in analyses but did not specify how. Could the authors please clarify? With such a wide spread of ages (5-year span), and given known variability in social patterns among adolescent females versus males, you would likely see significant variability in emotion regulation abilities as well as the types/severities/durations of teasing and internet harassment observed as a function of these demographic characteristics. These are critically important and should be included as covariates of interest in the analyses on all

  1. Were the bootstrapped confidence intervals used for interpreting indirect effects bias-corrected? Please clarify in the analytic strategy (I see it mentioned in the Results, but not in the Methods).

Results:

  1. Returning to comment 7 from the Methods, I see in Table 2 that age and sex do play an important role in explaining variance in both internet harassment and envy. These two variables should be more robustly included in the model and examined for their contributions to the entire process, not just controlled/held constant in the analytic strategy.

  1. Is the indirect effect reported a standardized or unstandardized value? Please specify.

  1. Could the authors clarify the moderation of the indirect effect (described on page 7, lines 274-278)? How is the full indirect effect (as opposed to just the “a” path) moderated in the model? Further, the parameter estimates provided don’t suggest any significant difference in the actual effects between high versus low ZYTS. This requires significant clarification.

Discussion:

  1. Overall, I understand that the authors are using the statistical significance of effects as their targets for interpretation, but I am very concerned about over-interpretation of effects. There is always a balance to be had between “statistical significance” and “practical significance”, and with such a simple model in such an enormous sample, even the smallest effects are likely to be statistically significant. I strongly encourage the authors to think critically about the value of some of these results (e.g., the standardized regression weights < .10) and how this might inform public policy, intervention, etc.

Minor points:

  1. The word “internet” is capitalized throughout the paper, please correct this.

  1. “Gender” should be “sex”.

  1. There are some typos and grammatical errors that need to be fixed.

  1. g., switching verb tenses in the results section
  2. Page 6 line 238, “the positive predictive effect of testing on internet harassment…”
  3. Page 10 line 393, “thus we cannot deduce casual relations.”

Author Response

Thank you very much for your kind comments on our manuscript (Manuscript reference No. ijerph-1646433). We have revised the manuscript according to the comments and below we address each of the reviewer’s comments with a point-by-point discussion describing the changes we have incorporated.

The following is our response to reviewers’ comments.

General Comments:

The present study explored the degree to which being teased was predictive of engagement in online harassment, and whether these associations were mediated by envy, and moderated by Zhong-Yong thinking style. The study uses questionnaire data from a large sample of youths, a major strength of the study. However, this is unfortunately where my enthusiasm for the paper dampens. Overall, the introduction and discussion sections offer some good insights but are rather choppy, leaving me wishing there was more fulsome, integrative discussion of the issues at large. Further, I am considerably worried about the primary outcome measure, harassment. The authors modified an existing questionnaire, which was designed to ask about the degree to which one has been harassed, and asked kids to instead indicate whether they have harassed others. This experience is extraordinarily subjective, and I’m concerned about the validity of such a measure. The tested models are overly simplistic and rely on simple overall mean scores obtained from questionnaires with no dissection of types/severity of teasing, envy, or harassment. Further, the analyses woefully overlook the importance of sex and age in an enormously dynamic period of development spanning the onset of puberty, a time in which these social behaviors are likely most seminal in a person’s life. Finally, the authors interpret extremely small effects as being quite meaningful (e.g., regression coefficients > .10). I urge the authors to consider both statistical and practical significance when interpreting their results. Even the most miniscule effects have a shot at being statistically significant when tested in a sample of 1800+ people, but that doesn’t make them useful or meaningful in the grander scope of understanding human behavior and mental wellness.

Response: Thanks very much for your important and detailed comments and suggestions, and below we addressed each of your specific comments point-by-point.

Specific Comments:

Introduction:

  1. I think that the intro could use some restructuring for the flow of information. The authors present the goals of the study at the end of the first section without defining several of the key components, e.g., envy and Zhong-Yong thinking style. The next section starts out with a clear definition of another component, teasing, which was only briefly described prior to introducing the study goals. Some basic reorganization of information would make the introduction much clearer and help the justification of the study.

Response: Many thanks for your suggestion. In the revised manuscript, we have reorganized some information of the introduction. First, we defined the variable of teasing at the end of the first section and the modified content is as follows: However, few studies have examined the relationship between teasing, a typical form of verbal bullying referring to a deliberate provocation accompanied by playful off-record markers that together comment on something of relevance to the target of the tease, and internet harassment perpetration. Hence, this study aims to investigate the impact of teasing on adolescent internet harassment perpetration and further explore the psychological mechanisms behind this relation.”. Moreover, we mentioned and defined key variable of envy at the beginning of “1.2. The mediating role of envy”, and the modified content is as follows: “For the aim of comprehending the “how” mechanism behind the relationship between teasing and Internet harassment, we investigated the mediating effect of envy on this association. Envy generally refers to a bitter emotion characterized by feelings of inferiority, hostility, and resentment, usually caused by a lack of others’ good qualities, achievements, or possessions in social comparisons”. Finally, the variable of Zhong-Yong thinking style was mentioned and defined at the beginning of “1.3. The moderating role of Zhong-Yong thinking style”. We hoped these basic reorganizations of information would help us present objectives and hypotheses of this research in a step-by-step manner and make the introduction much clearer.

  1. Please Figure 1 – the lines and text look distorted, and there are what appear to be paragraph marks next to the text in each of the boxes.

Response: Thanks for the comments. We have redone Figure 1.

Methods:

  1. Was there any rationale for the number of participants recruited? E.g., was this part of a larger study, was there any prior power analysis, etc.?

Response: Thanks for your comments. The participants were recruited from a middle school in Henan province of China using cluster random sampling. These participants were part of larger study on the experience of being bullied and social development among adolescents, so the number is relatively large.

  1. With so many participants, the authors have loads of statistical power to dissect these issues further and explore more nuanced indices of teasing and harassment (e.g., competency versus weight-related teasing). I would like to see more in-depth analyses pulling these issues apart and describing specifics. Lumping together all types of “teasing” and “harassment” is overly ambiguous, and there is no need to do so with such a large N

Response: Thanks very much for the important suggestions.

In the revised manuscript, we established two moderated mediation models with weight-related teasing and competency-related teasing as the independent variables respectively, envy as the mediating variable, and Zhong-Yong thinking style as the moderating variable. The following is the proposed moderated mediation model.

We used PROCESS 3.3 to examine the two moderated mediation models. The results indicated that weight-related/competency-related teasing was positively related to adolescent internet harassment. Mediation analyses revealed that envy partially mediated this relationship. Moderated mediation analyses further indicated that Zhong-Yong thinking style not only moderated the relationship between weight-related/competency-related teasing and envy, with the effect being stronger for adolescents with high levels of Zhong-Yong thinking, but also moderated the direct link between competency-related teasing and internet harassment, with the association being much more potent for adolescents with low levels of Zhong-Yong thinking. Nevertheless, Zhong-Yong thinking did not moderate the direct link between weight-related teasing and internet harassment.

As Zhong-Yong thinking style moderated the direct link between competency-related teasing and internet harassment, while did not moderate the direct link between weight-related teasing and internet harassment, corresponding explanations are also provided in the discussion section, and the revised content is as follows:

Regarding our hypothesis 4, the findings indicated Zhong-Yong thinking style buffered the relationship between competency-related teasing and internet harassment, whereas did not mitigate the relationship between weight-related teasing and internet harassment. That means the association between competency-related teasing and internet harassment was stronger for adolescents with low levels of Zhong-Yong thinking, while adolescents who has been teased about weight or body shape will perpetrate internet harassment regardless whether they have a high or low level of Zhong-Yong thinking.

On the one hand, competency is multidimensional, multiple thinking helps individuals assess their capabilities comprehensively from multiple perspectives. When adolescents are teased about abilities, such as academic performance or social skills, those holding high levels of Zhong-Yong thinking may not consider other’s negative opinion as a primary view of their competency. In addition, the model of hot/cool system (HCS) asserts that individuals will demonstrate more rational and self-control behavior in a cool system that is cognition-oriented [1]. Zhong-Yong thinking emphasizing rational cognition will active a cold system and allow the individual to respond appropriately, thereby inhibiting aggressive behavior [2]. Hence, competency-related teasing victims with high levels of Zhong-Yong thinking are less harmed by bad comments and choose more appropriate behaviors to respond to teasing, rather than internet harassment.

On the other hand, in today’s culture of “thinness is beautiful”, weight has become a sensitive topic and individuals have internalized stereotyped ideal body shape [3]. Adolescents are at an enormously dynamic period of self-concept development, juvenile victims of weight-related teasing tend to believe they are unattractive for not having an ideal body, forming negative self-image and low self-esteem [4,5]. Moreover, empirical research has revealed that weight-related teasing is more frequently experienced by adolescents overweight than healthy weight youth [6], whereas weight is difficult to change in a short period. Thus, the experience of being teased about weight can be a source of social pressure and much of a concern for youths, either holding high-level or low-level Zhong-Yong thinking. As a result, those victims are hard to behave rationally and are likely to vent their stress by perpetrating internet harassment.

References:

  1. Metcalfe, J.; Mischel, W. A hot/cool-system analysis of delay of gratification: Dynamics of willpower. Psychological Review 1999, 106, 3.
  2. Wei, H.; Ding, H.; Huang, F.; Zhu, L.; Addiction. Parents’ phubbing and cyberbullying perpetration among adolescents: The mediation of anxiety and the moderation of Zhong-Yong Thinking. International Journal of Mental Health 2021, 1-14.
  3. Chen, G.; Guo, G.; Wu, S.; Zhou, L.; Xiao, S.; Cai, T. Effect of weight-related teasing on dieting in a sample of high school students: Mediating effect of body dissatisfaction and moderating effect of gender. Chinese Journal of Clinical Psychology 2019, 27, 108-112.
  4. Chen, G.; He, J.; Zhang, B.; Fan, X. Body weight and body dissatisfaction among Chinese adolescents: Mediating and moderating roles of weight-related teasing. Current Psychology 2019, 1-9.
  5. Schaefer, M.K.; Salafia, E.H.B. The connection of teasing by parents, siblings, and peers with girls' body dissatisfaction and boys' drive for muscularity: The role of social comparison as a mediator. Eating Behaviors 2014, 15, 599-608.
  6. Neumark-Sztainer, D.; Falkner, N.; Story, M.; Perry, C.; Hannan, P.J.; Mulert, S. Weight-teasing among adolescents: correlations with weight status and disordered eating behaviors. International Journal of Obesity 2002, 26, 123-131.

  1. Could the authors more clearly identify how the Facebook Envy Scale was adapted to reflect Chinese social media sites? Also, how does social media usage/participation/consumption in China compare to that of other nations where social media sites are not state-controlled/limited?

Response: Many thanks for your comments. The current study adopted the Facebook Envy Scale to evaluate the levels of envy. Due to specific national condition of China, our netizens cannot use Facebook, but there are social network sites with similar shape and function as Facebook, such as WeChat, QQ, and Weibo. Given the utilization rates of WeChat, QQ, and Weibo reached 83.4%, 58.8%, and 42.3% respectively, becoming the three mainstream social network sites in China at the time of this study [1], the Facebook context was changed to the contexts of WeChat, Weibo and QQ in the current study. In addition, previous studies in Chinese culture demonstrated that this measure has good reliability [2,3]. In this study, the Cronbach’s α was 0.82.

References:

  1. CNNIC (2019). The 43th Statistical Report on China’s Internet Development
  2. Ding, Q.; Zhang, Y.-X.; Wei, H.; Huang, F.; Zhou, Z.-K. Passive social network site use and subjective well-being among Chinese university students: A moderated mediation model of envy and gender. Personality and Individual Differences 2017, 113, 142-146.
  3. Lian, S.; Sun, X.; Niu, G.; Zhou, Z. Upward social comparison on SNS and depression: A moderated mediation model and gender difference. Acta Psychologica Sinica 2017.

  1. Why did the authors choose to use averages of item scores for each scale as their primary measures, as opposed to other metrics that better capture the variability in scales (e.g., sum scores, factor scores, etc.)?

Response: Thanks for the important comments. In the revised manuscript, we have chosen to use sum scores of each scale as our primary measures instead of using average scores, and updated the relevant table, figure and content in the method and result section.

  1. I’m concerned about the validity of the internet harassment measure in its modified form for the current study. Asking youths if someone has made offensive comments toward them is likely to receive more honesty/accuracy than asking youths if they have said something offensive to another person. Further, perceptions of what is “offensive” versus “a joke/funny” are likely to vary significantly from person to person. Thus, youths engaging in harassment might not answer these questions accurately with respect to the recipient’s perceptions of the situation (e.g., the perpetrator thought it was just a funny joke, whereas the recipient saw it as offensive/hurtful). The authors need to strongly justify and defend the validity of this scale.

Response: Thanks for the comments. Firstly, we found that previous researches measured internet harassment perpetration by querying if youth had engaged in two possible internet harassment behaviors in the past year: (1) making rude or nasty comments to someone on the internet, and (2) using the internet to harass or embarrass someone with whom the youth was mad [1,2]. This measurement content is similar to ours that ask youths whether they made offensive or threatening comments to others online in the past year. Secondly, the Cronbach’s α coefficient for the present sample was 0.85, demonstrating the scale has very good reliability. Finally, empirical research has shown that the perpetrator might underestimate bullying rates, whereas victims might overestimate it, due to using self-reported questionnaires [3]. Therefore, in the present study using self-assessment, adolescents engaging in harassment might not answer these questions accurately. This is exactly one of limitations of our study and future research can focus as we modified in the “4.4. Limitations and implications” of the discussion part. The modified content is as follows: “Some limitations should be acknowledged in the current study. First, we only used self-reported measures to collect data, social desirability effects or response bias may exist in this work. For example, adolescents are likely to underestimate their internet harassment perpetration. Future research should improve the objectivity of data by collecting from a variety of sources, such as parents, peers, or school administrators”.

References:

  1. Ybarra, M.L.; Mitchell, K.J.J.J.o.a. Youth engaging in online harassment: Associations with caregiver–child relationships, Internet use, and personal characteristics. 2004, 27, 319-336.
  2. Ybarra, M.L.; Mitchell, K.J.J.J.o.A.H. Prevalence and frequency of Internet harassment instigation: Implications for adolescent health. 2007, 41, 189-195.
  3. Bouman, T.; van der Meulen, M.; Goossens, F.A.; Olthof, T.; Vermande, M.M.; Aleva, E.A.J.J.o.s.p. Peer and self-reports of victimization and bullying: Their differential association with internalizing problems and social adjustment. 2012, 50, 759-774.

  1. Please specify the version of PROCESS used for analyses.

Response: Many thanks for your suggestion. We used PROCESS 3.3 to analyze data, and specified the version of PROCESS in “2.4. Data analysis”.

  1. The authors state that they controlled for age and sex in analyses but did not specify how. Could the authors please clarify? With such a wide spread of ages (5-year span), and given known variability in social patterns among adolescent females versus males, you would likely see significant variability in emotion regulation abilities as well as the types/severities/durations of teasing and internet harassment observed as a function of these demographic characteristics. These are critically important and should be included as covariates of interest in the analyses on all.

Response: Thanks very much for your sincere suggestions.

Firstly, the aims of the present study are to investigate the impact of weight-related/competency-related teasing on adolescent internet harassment perpetration, and further explore the mediating role of envy and the moderating role of Zhong-Yong thinking style. The main study variables are weight-related/competency-related teasing, envy, internet harassment, and Zhong-Yong thinking style, instead of sex and age.

Secondly, compared with the experiment, there are fewer variables controlled for in the questionnaire, which was compensated for by using statistical methods of control [1]. In the current study, we found sex had a significantly negative relationship with envy, while positively associated with internet harassment. Age was positively linked with both envy and internet harassment. Referring to previous studies [2,3], we chose to include sex and age as covariates in analyses to obtain pure mediating and moderating effects. Specifically, sex and age were entered into the regression model along with the main study variables. For example, in order to explore if weight-related teasing and envy can significantly predict internet harassment, the variables of sex, age, weight-related teasing, and envy will be entered into the regression equation together.

Finally, due to sex and age may play an important role in explaining variance in both envy and internet harassment, we added some content in “4.4. Limitations and implications”. The added content is as follows: “Finally, the results of correlation analysis showed that females were more likely to envy and perpetrated less internet harassment compare to males. In addition, teenagers are more inclined to experience envy and commit more internet harassment with the growth of age. However, we included sex and age as covariates in analyses to obtain pure mediating and moderating effects. Future studies should further explore the vital role of sex and age in explaining variance in both envy and internet harassment, or investigate whether sex and age moderate the mediating effect of envy”.

References:

  1. Wen, Z. Causal inference and analysis in empirical studies. Journal of Psychological Science 2017, 40, 200-208.
  2. Wang, P.; Liu, S.; Zhao, M.; Yang, X.; Zhang, G.; Chu, X.; Wang, X.; Zeng, P.; Lei, L. How is problematic smartphone use related to adolescent depression? A moderated mediation analysis. Children Youth and Services Review 2019, 104, 104384.
  3. Wang, W.; Xie, X.; Wang, X.; Lei, L.; Hu, Q.; Jiang, S. Cyberbullying and depression among Chinese college students: A moderated mediation model of social anxiety and neuroticism. Journal of Affective Disorders 2019, 256, 54-61.

  1. Were the bootstrapped confidence intervals used for interpreting indirect effects bias-corrected? Please clarify in the analytic strategy (I see it mentioned in the Results, but not in the Methods).

Response: Many thanks for your comments. In the revised manuscript, we have clarified the method used to test the mediation and moderating effects in “2.4. Data analysis”. The modifications are as follows: “The mediation and moderating effects were tested by using the bias-corrected percentile Bootstrap method (5000 bootstrap samples) with 95% confidence intervals (CI). If the 95% confidence interval excludes zero, it means that the effects are statistically significant”.

Results:

  1. Returning to comment 7 from the Methods, I see in Table 2 that age and sex do play an important role in explaining variance in both internet harassment and envy. These two variables should be more robustly included in the model and examined for their contributions to the entire process, not just controlled/held constant in the analytic strategy.

Response: Thanks for your comments. Table 2 presented the results of mediation testing, demonstrating envy played a mediating role. Because age and sex do play an important role in explaining variance in both internet harassment and envy, it is necessary to include sex and age as covariates in analyses to obtain pure mediating and moderating effects. As the response to comment 7 from the Methods, sex and age were entered into the regression equation along with the main study variables to test the moderated mediation models.

  1. Is the indirect effect reported a standardized or unstandardized value? Please specify.

Response: Thanks for your comments. The indirect effect value we reported was standardized, and we specified it in “3.2. Testing for mediation effect”.

  1. Could the authors clarify the moderation of the indirect effect (described on page 7, lines 274-278)? How is the full indirect effect (as opposed to just the “a” path) moderated in the model? Further, the parameter estimates provided don’t suggest any significant difference in the actual effects between high versus low ZYTS. This requires significant clarification.

Response: Thanks very much for your sincere comments. After careful consideration, we have decided to remove this section for the following two reasons. First, PROCESS 3.3 didn’t provide difference test of indirect effect between high versus low ZYTS. Second, one of the aims of our research was to examine whether ZYTS moderate the direct association between weight-related/competency-related teasing and internet harassment (“a” path), rather than test whether the full indirect effect is moderated by ZYTS. Hence, out of caution and for the purposes of this study, we deleted this content. 

Discussion:

  1. Overall, I understand that the authors are using the statistical significance of effects as their targets for interpretation, but I am very concerned about over-interpretation of effects. There is always a balance to be had between “statistical significance” and “practical significance”, and with such a simple model in such an enormous sample, even the smallest effects are likely to be statistically significant. I strongly encourage the authors to think critically about the value of some of these results (e.g., the standardized regression weights < .10) and how this might inform public policy, intervention, etc.

Response: Thanks very much for your comments.

Previous research has indicated that the moderating effect and moderated mediation effects are usually small, but the corresponding studies are often still relevant in the research of psychology and behavior [1]. Firstly, in non-experimental studies, it is indisputable that the effect size is small [2]. Secondly, if a small amount of effect can lead to important outcomes (e.g., internet harassment), if it can accumulate over time, and if it can have a significant impact on the lives of many young people, it can be practically important [3,4].

In this study, although the smallest effects are likely to be statistically significant with an enormous sample, the findings will be more accurate and credible with a larger and representative sample size. Moreover, our findings are in line with previous theories and empirical researches, indicating weight-related/competency-related teasing is a risk factor for internet harassment perpetration, and envy plays a mediating role as well as ZYTS plays a moderating role in this relationship among many youths. Whether weight-related/competency-related teasing, envy, ZYTS, or internet harassment are all vital factors for the development of adolescents. Therefore, our results are practical to some extent.

References:

  1. Ye, B.; Wen, Z. A discussion on testing methods for mediated moderation models: discrimination and integration. Acta Psychologica Sinica. 2013, 45, 1050-1060.
  2. Aguinis, H.; Beaty, J.C.; Boik, R.J.; Pierce, C.A. Effect size and power in assessing moderating effects of categorical variables using multiple regression: A 30-year review. Journal of Applied Psychology. 2005, 90, 94.
  3. Ellis, P.D. The essential guide to effect sizes: Statistical power, meta-analysis, and the interpretation of research results; Cambridge university press: 2010.
  4. Prentice, D. A., & Miller, D. T. (2016). When small effects are impressive. In A. E. Kazdin (Ed.), Methodological issues and strategies in clinical research (pp. 99–105). American Psychological Association

Minor points:

  1. The word “internet” is capitalized throughout the paper, please correct this.

Response: Thanks for the comments. We have corrected this mistake.

  1. “Gender” should be “sex”.

Response: Many thanks for the comments. We have corrected this mistake.

There are some typos and grammatical errors that need to be fixed.

g., switching verb tenses in the results section

Page 6 line 238, “the positive predictive effect of testing on internet harassment…”

Page 10 line 393, “thus we cannot deduce casual relations.”

Response: Thanks for the important comments. We have carefully reviewed our manuscript and corrected some typos and grammatical errors.

We thank you and the reviewers for the comments which make us think and learn a lot, and believe that the incorporation of the amendments makes this manuscript more valuable. We hope the manuscript is now suitable in its present form for publication in International Journal of Environmental Research and Public Health.

Wish you all the best!

Yours sincerely,

Authors

Reviewer 3 Report

Please, check the language - it needs a revision as there are letters missing from words. Also, ask a native speaker to make fine changes

Do not use WOULD in a hypothesis

Re-do Figure 1 - it has pause marks 

There are a lot of phrases used similarly to other papers, especially in the research methodology, please rephrase them using your own words. I have attached the plagiarism report.

Author Response

Thank you very much for your kind comments on our manuscript (Manuscript reference No. ijerph-1646433). We have revised the manuscript according to the comments and below we address each of the reviewer’s comments with a point-by-point discussion describing the changes we have incorporated.

The following is our response to reviewers’ comments.

  1. Please, check the language - it needs a revision as there are letters missing from words. Also, ask a native speaker to make fine changes.

Response: Thanks for the important suggestions. We have revised our manuscript and asked a native speaker to make fine changes.

  1. Do not use WOULD in a hypothesis

Response: Many thanks for the comments. We have corrected this mistake.

  1. Re-do Figure 1 - it has pause marks

Response: Thanks for the comments. We have redone Figure 1.

  1. There are a lot of phrases used similarly to other papers, especially in the research methodology, please rephrase them using your own words. I have attached the plagiarism report.

Response: Many thanks for the suggestion. We have rephrased some sentences according to the plagiarism report, especially in the research methodology.

We thank you and the reviewers for the comments which make us think and learn a lot, and believe that the incorporation of the amendments makes this manuscript more valuable. We hope the manuscript is now suitable in its present form for publication in International Journal of Environmental Research and Public Health.

Wish you all the best!

Yours sincerely,

Authors

Round 2

Reviewer 2 Report

I would like to commend the authors for their efforts in revising this manuscript. I appreciate the attention to detail in their responses, and in the subsequent adjustments to the text. I do still think that this paper would greatly benefit from an in-depth dive into age- and sex-based effects given the known differences in social experience between male and female youths, particularly in this wide age rage (11-16 years-old), but I will leave it to the editor to decide if the paper is fulsome enough without these additional analyses. Otherwise, I think that the paper could use a bit more editing for grammatical inconsistencies.

Author Response

Thank you very much for your kind comments on our manuscript. We have revised the manuscript according to the comments and below we addressed each of the reviewer’s comments with a point-by-point discussion describing the changes we have incorporated.

The following is our response to reviewers’ comments.

Comments and Suggestions for Authors:

I would like to commend the authors for their efforts in revising this manuscript. I appreciate the attention to detail in their responses, and in the subsequent adjustments to the text. I do still think that this paper would greatly benefit from an in-depth dive into age- and sex-based effects given the known differences in social experience between male and female youths, particularly in this wide age rage (11-16 years-old), but I will leave it to the editor to decide if the paper is fulsome enough without these additional analyses. Otherwise, I think that the paper could use a bit more editing for grammatical inconsistencies.

Response: Thanks very much for your sincere suggestions.

In the revised manuscript, we conducted additional analyses regarding age and sex, and clarified the corresponding results and discussions. The added content in “3.1. Preliminary analysis” is as follows: “Notably, we divided the participants into early adolescence (11-13 years old) and middle adolescence (14-16 years old) based on their development trajectory and age range. Then we conducted independent-sample t test for age as well as sex. The results revealed that there were no significant age and gender differences in weight-related teasing, competency-related teasing, and Zhong-Yong thinking style, whereas both age and sex differences existed in envy and internet harassment. Specifically, the levels of envy (t = 2.25, p < 0.05) and internet harassment (t = 2.55, p < 0.05) were significantly higher in middle adolescence than in early adolescence. Besides, females reported significantly higher envy scores (t = 2.34, p < 0.05) and lower internet harassment scores (t = − 3.93, p < 0.001) than males”. In the section of “4. Discussion”, we added the following content: “It is worth emphasizing that the results of the age difference test revealed that from early to middle adolescence, individuals had significantly higher levels of envy and committed more internet harassment. This may be because as teenagers get older, they are more likely to have access to the internet and spend more time on social media sites. When youths notice some information in terms of other users’ happy smiles, wonderful life, and excellent achievements on social network sites, they will become envious [1]. Additionally, as more and more people engage in cyber aggression, adolescents using social network sites will inevitably notice and then imitate those deviant behaviors, resulting in internet harassment. The sex difference test also showed that females suffered more feelings of envy than males, and perpetrates less internet harassment. These findings are consistent with the notion that significant variability in emotion regulation abilities as well as the severities of internet harassment perpetration exist among adolescent females versus males [2,3]. On the on hand, compared with males, females are more sensitive to comparisons with others, and more frequently ruminate on their inferiority, then generating more feelings of envy. On the other hand, males have lower empathy than girls, while people with low empathy are more inclined to engage in internet harassment [4,5]”.

References:

  1. Tandoc Jr, E.C.; Ferrucci, P.; Duffy, M. Facebook use, envy, and depression among college students: Is facebooking depressing? Computers in Human Behavior 2015, 43, 139-146.
  2. Nolen-Hoeksema, S. Emotion regulation and psychopathology: The role of gender. Annual Review of Clinical Psychology 2012, 8, 161-187.
  3. Wang, B.; Jin, C.; Zhao, B. Relationships among family function, interpersonal adaptation and cyberbullying of adolescents: A moderated mediation effect. Psychological Development and Education 2020, 36, 469-476.
  4. Jolliffe, D.; Farrington, D.P. Development and validation of the Basic Empathy Scale. Journal of Adolescence 2006, 29, 589-611.
  5. Kokkinos, C.M.; Kipritsi, E. Bullying, moral disengagement and empathy: Exploring the links among early adolescents. Educational Psychology 2018, 38, 535-552.

We thank you and the reviewers for the comments which make us think and learn a lot, and believe that the incorporation of the amendments makes this manuscript more valuable. We hope the manuscript is now suitable in its present form for publication in International Journal of Environmental Research and Public Health.

Wish you all the best!

Yours sincerely,

Authors

Reviewer 3 Report

.

Author Response

Thank you very much for your kind comments on our manuscript. We have revised the manuscript according to the comments and below we addressed each of the reviewer’s comments with a point-by-point discussion describing the changes we have incorporated.

The following is our response to reviewers’ comments.

Response: Many thanks for you. Your important comments and suggestions have made a great contribution to our article.

We thank you and the reviewers for the comments which make us think and learn a lot, and believe that the incorporation of the amendments makes this manuscript more valuable. We hope the manuscript is now suitable in its present form for publication in International Journal of Environmental Research and Public Health.

Wish you all the best!

Yours sincerely,

Authors
